# A Dynamic Probabilistic Model for Heterogeneous Data Fusion: A Pilot Case Study from Computer-Aided Detection of Depression

**DOI:** 10.3390/brainsci13091339

**Published:** 2023-09-18

**Authors:** Federica Vitale, Bruno Carbonaro, Anna Esposito

**Affiliations:** 1Department of Mathematics and Physics, University of Campania “L. Vanvitelli”, Viale Lincoln 5, 81100 Caserta, Italybruno.carbonaro@unicampania.it (B.C.); 2Department of Psychology, Università degli Studi della Campania “L. Vanvitelli”, Viale Ellittico 31, 81100 Caserta, Italy; 3International Institute for Advanced Scientific Studies (IIASS), 84019 Vietri sul Mare, Italy

**Keywords:** probability, kinetic theory, mathematical models, speech recognition, statistical parametric speech synthesis

## Abstract

The present paper, in the framework of a search for a computer-aided method to detect depression, deals with experimental data of various types, with their correlation, and with the way relevant information about depression delivered by different sets of data can be fused to build a unique body of knowledge about individuals’ mental states facilitating the diagnosis and its accuracy. To this aim, it suggests the use of a recently introduced «limiting form» of the kinetic-theoretic language, at present widely used to describe complex systems of objects of the most diverse nature. In this connection, the paper mainly aims to show how a wide range of experimental procedures can be described as examples of this «limiting case» and possibly rendered by this description more effective as methods of prediction from experience. In particular, the paper contains a simple, preliminary application of the method to the detection of depression, to show how the consideration of statistical parameters connected with the analysis of speech can modify, at least in a stochastic sense, each diagnosis of depression delivered by the Beck Depression Inventory (BDI-II).

## 1. Introduction

This paper deals with a special but important and interesting case of the problem of treating experimental data of various types (*heterogeneous* data) together, to provide a detailed description of a subject, or in particular to classify it with a property chosen *a priori*.

The starting point of the work is finding methods and tools for implementing computer-aided devices for the detection of major depressive disorders (MDD), i.e., the automatic classification of any given subject with respect to a set of previously identified possible depressive states. This task requires in general the analysis of a rather large amount of data obtained from experiments of various kinds and settled for investigating whether features describing the collected data can discriminate between healthy and depressed participants. The involved experiments aimed to gather behavioral data (such as speech, handwriting, facial, vocal, and gestural expressions) from healthy/depressed diagnosed subjects by defining behavioral tasks known to be affected by depressive states. The collected data were then appropriately processed to extract features describing changes in the healthy perception of individuals due to depressive states.

Depression is classified as a mood disorder by the Diagnostic and Statistical Manual of Mental Disorders (DSM-5, American Psychiatric Association, 2013). The DSM-5 report as its symptoms the impairment impaired of emotional expressions, cognitive functions, social behavior, social relationships, and physical functioning (changes in appetite or in weight, headaches, pain). However, not all depressed individuals meet these criteria. Some individuals may show irritable rather than depressed mood and may not be classified as depressed. “The conditions and underlying physiologies associated with the experience of depression may vary widely, and no core composition can be assumed” ([1], p. 1). Given the complexity of the accompanying symptoms, it was shown that the automatic analysis of behavioral data can provide quantitative measures for describing changes affecting healthy/depressed individuals (see [2] for depression symptoms and [3] and reference therein for automatic behavioral analyses) calling for the identification of a multi-dimensional mathematical model serving for the implementation of automated and cost-effective technological systems devoted to the early detection of depressive states.

The mathematical model proposed in this paper has been applied to speech data. More precisely, in this case, for any subject, the experimental procedure involves the collection of speech data obtained from the reading of a narrative text and the free description of daily activities performed during a week, and the answers provided by the participants to data collection to the questions of BDI-II [4]. Further data can be collected on this line of conduct, other than speech, including facial expressions, handwriting and drawing, body movements, hands’ gestures, and any other activity useful for behavior analysis ([5,6,7,8,9,10,11,12,13], among others). Though all the outcomes of these experiments can be (and actually are) given a quantitative expression, by means of suitably defined measures or scores, nevertheless, this raises a new problem, regarding the comparison of heterogeneous measures.

To solve this problem, a good method seems to be shifting the attention from the measures as such to their joint probabilities and, as a consequence, to their mutual conditional probabilities. This goal has been achieved by the use of a recently introduced «limiting form» of the kinetic-theoretic language [14], a *scheme* of models based on the use of a generalization of Boltzmann equation governing gas dynamics [15,16], which in the last fifty years has been proposed to describe the evolution of many-particle systems in which the interactions between elements go far beyond purely mechanical ones, and can produce unexpected and not deterministically predictable behaviors, both mechanical and non-mechanical. The elements of the system are themselves no longer just particles in the mechanical sense, but individuals of the most varied nature: cells [17,18,19,20,21], cars [22,23], insects (dynamics of swarms) [24], and finally, human beings and possibly care structures [21,25,26,27,28,29].

The key idea the whole scheme relies upon is that the results of pairwise interactions between the «particles» of the system cannot be predicted with certainty, but they still produce a *deterministic* evolution of the (relative) frequency (or, in appropriate contexts, probability) distribution on the set (space) of possible states of particles [15,30].

One of the techniques often employed in the analysis of the dependence of this distribution on the interactions consists in subdividing the system into *homogeneous* subsystems (i.e., containing the same kind of objects), which can also have different spaces of states. No models, however, have been outlined in which at least one functional subsystem consisted of abstract objects, e.g., «procedures» or «experiments» or «diagnostic methods».

In addition, as is quite natural, in all models based on the Kinetic Theory, the time variable is continuous, and as a consequence the evolution of the system turns out to be governed by a system of differential equations, when the space D of all possible states of the «particles» is discrete (see Section 2) or of integro-differential equations, when D is continuous, and described by either functions of time or functions of state variables and time. In this paper, a formulation of the model introduced in [14] is used, in which the complex system under examination contains such abstract objects as «random variables» and the time variable is allowed to be *discrete*.

The proposed model is applied to speech data, collected from depressed (diagnosed from medical doctors) and healthy subjects. The research assumes that in the acoustic and linguistic content of people’s speeches it is possible to capture information regarding their psychological state, according to the narrative psychological content analysis framework [31].

There are several studies showing that there are vocal changes in the speech of subjects with depression. These changes concern, mainly, (a) the fundamental frequency f0, which seems to reduce its frequency range and average values in the acute stages of the pathology; (b) the speaking rate which seems to increase as the pathology improves, and finally, that this improvement is related to (c) a decrease in the number and duration of speech pauses (for a review, see [3,32]).

The data utilized for the following research included 24 subjects divided into an experimental and control group. The experimental group, included 12 subjects (nine females) with severe (four subjects), moderate (five) and mild (three) depression, aged from 27 to 60 years (mean age = 41.07 years; standard deviation = 12.5). The control group was selected to match the experimental groups for age and gender.

The diagnostic criteria of the Beck Depression Inventory, Second Edition (BDI II, [4]), an instrument for self-assessment of depression severity in already diagnosed patients and for detecting the risk of depression in the normal population, were used to assess depression. The BDI consists of 21 sets of statements to be rated on a four-point scale. A score of >16 indicates the presence of depression in the patient.

Subjects with depression were recruited in Italy, from the mental health department of the ASL of Caserta, the mental hygiene institute of the ASL of Santamaria Capua Vetere, the Psychological Listening Center of Aversa, and a private medical practice.

The task performed by the subjects, divided into two parts, consisted of a reading of the Aesop’s tale “the North Wind and the Sun” visualized on a computer screen (see Figure 1).

In the second part, participants were required to produce a spontaneous narrative (diary), describing salient events which happened during the previous week, highs and lows with friends, relatives, co-workers and also, in many cases, opinions or criticisms of the Italian political situation.

During the reading and diary sections, the subjects were audio and/or video recorded upon consent. The duration of the recordings varied. The diary ranges from a minimum of 2 min to a maximum of 5/6 min; while the reading takes about 50 s. The total time of the task, from meeting with the examiner to the end of the recording, was around 15 min.

The recordings were made using a clip-on microphone (Audio-Technica ATR3350), with an external USB sound card. Speech was sampled at 16 kHz and quantized at 16 bits.

This software/hardware was developed inside the European Space Agency (ESA 2012) funded project (n. HSO-US 2012-108) “Psychological Status Monitoring by Computerised Analysis of Language phenomena (COALA)” of which one of the authors (Professor Anna Esposito) was the coordinator. It utilized standard software/hardware equipped with a sound card to sample the speech data. It can run on any operating system. It was installed on Window 7 for collecting the COALA data. In the following years it has been installed on the operating systems Windows 8, 9 and 10 and can also be installed on Windows 11. The data exploited in this study were collected between years 2016 and 2017, after obtaining the permission of the ethical committee of the department of psychology of the Università della Campania “Luigi Vanvitelli”. The date displayed in Figure 1 and the time in which the data were collected do not affect the originality and innovation of the results presented in this study, which seeks to propose a mathematical model to fuse heterogeneous data.

The recording took place in an isolated and soundproofed room, granted by the health center where the data collection was taking place.

Prior to the recording, subjects signed an informed consent form for data processing and filled out the BDI II [33], while the examiner prepared the instrumentation consisting of a personal computer and recording software.

The research had received the approval from the ethical committee of the department of Psychology, University of Campania “Luigi Vanvitelli” with the protocol code 09/2016.

The model suggests a way to give a sharp definition of states and a precise method to place each individual in a state and to confirm or modify the placement after each «interaction». This point will be discussed in details in the following Sections. More precisely, in Section 2 and Section 3, the formulation of the «discretized» model is presented, in Section 4, Section 5 and Section 6, the language to the detection of depression using the results reported in [34] is applied, and in Section 7 some possible research perspectives are outlined.

## 2. The Standard Model

In order to provide the reader at least some hints about the origin of the methods that will be used in the next Sections to analyze the correlation between different features possibly describing depression, the current one is devoted to recall the main features of the kinetic-theoretic language in its most general form.

Our starting point is a set *S* of a very large number of objects of any nature (cells, living individuals, and also—as was shown in [21] and will be discussed in more details in the next Section—care structures, e.g., hospitals and systems of care tools and experimental devices) called *individuals* or *(active) particles*. In most cases of interest, *S* is assumed to be the joining of a family $\left\{S_{1},S_{2},\;\ldots \;,S_{k}\right\}$ of subsystems, called its *functional subsystems*. The introduction of such subsystems finds its natural application in biology, for instance to describe the fight between tumor cells and immune system, or the competition between different species to model Darwinian selection, or in social sciences, to model interactions between social classes, or in economics, to model the fluxes of wealth. Accordingly, for each subsystem $S_{i}$, the *state* of each individual in $S_{i}$ is defined according to the context in which the present description should be developed, and can be expressed as a scalar variable $u_{i}$ or a vector variable $u_{i}\equiv \left(u_{i}^{1},u_{i}^{2},\;\ldots \;,u_{i}^{m}\right)$ which will be called *state variable*. This variable may describe any property we can find suitable for our research according to the context (biological, economic, social, etc.) in which it is performed. According to the context, the state variable may be assumed to take its values in a *discrete* domain or in a *continuous* domain: in the former case, the domain can be a finite or countable subset of Z or $Z^{m}$, while in the latter case it can be a bounded or unbounded real interval or a bounded or unbounded domain of $R^{m}$. In both cases, the domain $D_{u_{i}}$ (or $D_{u_{i}}$) of the state variable is called the *state space* of subsystem $S_{i}$ (in some cases, the same variable can be used to identify the state of the members of all the functional subsystems, but in most cases different state variables must be used for different subsystems).

As in Boltzmann’s Kinetic Theory of Gases, the mathematical framework developed in [21] is statistical, i.e., one is interested to describe the evolution in time of the system as a whole rather than the evolution of the states of single particles. More precisely, it must be acknowledged that any precise description of the state of each particle of *S* must be given up, for both theoretical and practical reasons [16,35], so one cannot but decide to study the probability (expressed experimentally in terms of relative frequency) distribution over the state space at each instant. This means that the state variable is conceived as a *random variable* at each instant and the aim of the study is the forecast of its probability density function at any time.

In this Section, with no loss of generality, and only for the sake of simplicity and in view of the application to be discussed in the sequel, it will be described here in some detail only the case of a system *S* consisting in two different subsystems $S_{1}$ and $S_{2}$ such that the state of individuals in each $S_{h}$ is described by a different *discrete* scalar variable $u_{h}$. We set
$$D_{h}=\left\{u_{h,1},u_{h,2},\;\ldots \;,u_{h,m_{h}}\right\}\;\left(h=1,2\right)\;.$$

For any *t* in a time-interval I⊆R, the state of system $S_{h}$ at time *t* will be identified by a probability distribution (also called *state vector*) $f_{t}^{\left(h\right)}\equiv \;(f_{t}^{\left(h\right)}\left(u_{h,1}\right),\;f_{t}^{\left(h\right)}\left(u_{h,2}\right)$, …, $f_{t}^{\left(h\right)}\left(u_{h,m_{h}}\right))$ on $D_{h}$, and, for any $k\in \{1,2,\;\ldots \;,m_{h}\}$, one can define the function $f_{k}^{\left(h\right)}\;:\;t\in I\;⟶\;f_{k}^{\left(h\right)}\left(t\right)\equiv \;f_{t}^{\left(h\right)}\left(u_{h,k}\right)\in \left[0,1\right]$. According to this definition, one has
$$\sum_{k=1}^{m_{h}}f_{k}^{\left(h\right)}\left(t\right)=\sum_{k=1}^{m_{h}}f_{t}^{\left(h\right)}\left(u_{h,k}\right)=1\;\left(h=1,2\right)\;.$$

Now, *if* the evolution of the system *S* is envisaged as a time-continuous stochastic process, then the time derivative of each probability function $f_{k}^{\left(h\right)}$ is expressed, according to the law of alternatives, in terms of transition probabilities. More precisely,

1.for any $\left(r,s,j\right)\in {\left\{1,2,\;\ldots \;m_{h}\right\}}^{3}$, the symbol $F^{r}{}_{sj}\left(h\right)\equiv F_{h}\left(u_{h,s},u_{h,j};u_{h,r}\right)$ will denote the probability that a particle of $S_{h}$
*falls* from the state $u_{h,s}$ to the state $u_{h,r}$ after an interaction with another particle of $S_{h}$ which is in the state $u_{h,j}$: accordingly,
$$\sum_{r=1}^{m_{h}}F^{r}{}_{sj}\left(h\right)=1\;,\;\forall \;\left(s,j\right)\in {\left\{1,2,\;\ldots \;,m_{h}\right\}}^{2}\;\left(h=1,2\right)\;.$$2.for any $\left(r,s\right)\in {\left\{1,2,\;\ldots \;m_{h}\right\}}^{2}$, and for any $j\in \{1,2,\;\ldots \;,m_{k}\}$ (with k≠h), the symbol $\Phi^{r}{}_{sj}\left(h,k\right)\equiv \Phi_{hk}\left(u_{h,r},u_{k,j};u_{h,s}\right)$ will denote the probability that a particle of $S_{h}$
*falls* from the state $u_{h,s}$ to the state $u_{h,r}$ after an interaction with a particle of $S_{k}$ which is in the state $u_{k,j}$: accordingly,
$$\begin{array}{cc} \sum_{r=1}^{m_{h}}\Phi^{r}{}_{sj}\left(h,k\right)=1\;,\;\forall \;s\in \left\{1,2,\;\ldots \;,m_{h}\right\}\;,\; & \forall j\in \{1,2,\;\ldots \;,m_{k}\}\;,\\ & (h,k=1,2\;;\;k\neq h)\;. \end{array}$$

So, the law of alternatives yields the following system of differential equations:(1)$$\begin{array}{cc} \; & \frac{df_{h}^{\left(1\right)}}{dt}\left(t\right)=\sum_{i,j}^{1\ldots m_{1}}\tau_{ij}^{1}F^{h}{}_{ij}\left(1\right)f_{i}^{\left(1\right)}\left(t\right)f_{j}^{\left(1\right)}\left(t\right)-\sum_{i,j}^{1\ldots m_{1}}\tau_{hj}^{1}F^{i}{}_{hj}\left(1\right)f_{h}^{\left(1\right)}\left(t\right)f_{j}^{\left(1\right)}\left(t\right)+\\ & \;+\sum_{i=1}^{m_{1}}\sum_{j=1}^{m_{2}}\left[\eta_{ij}^{12}\Phi^{h}{}_{ij}\left(1,2\right)f_{i}^{\left(1\right)}\left(t\right)f_{j}^{\left(2\right)}\left(t\right)-\eta_{hj}^{12}\Phi^{i}{}_{hj}\left(1,2\right)f_{h}^{\left(1\right)}\left(t\right)f_{j}^{\left(2\right)}\left(t\right)\right],\;\left(h=1,\;\ldots \;,m_{1}\right)\\ \; & \frac{df_{h}^{\left(2\right)}}{dt}\left(t\right)=\sum_{i,j}^{1\ldots m_{2}}\tau_{ij}^{2}F^{h}{}_{ij}\left(2\right)f_{i}^{\left(2\right)}\left(t\right)f_{j}^{\left(2\right)}\left(t\right)-\sum_{i,j}^{1\ldots m_{2}}\tau_{hj}^{2}F_{hj}^{i}\left(2\right)f_{h}^{\left(2\right)}\left(t\right)f_{j}^{\left(2\right)}\left(t\right)+\\ & \;+\sum_{i=1}^{m_{2}}\sum_{j=1}^{m_{1}}\left[\eta_{ij}^{21}\Phi^{h}{}_{ij}\left(2,1\right)f_{i}^{\left(2\right)}\left(t\right)f_{j}^{\left(1\right)}\left(t\right)-\eta_{hj}^{21}\Phi_{hj}^{i}\left(2,1\right)f_{h}^{\left(2\right)}\left(t\right)f_{j}^{\left(1\right)}\left(t\right)\right],\;\left(h=1,\;\ldots \;,m_{2}\right) \end{array}$$
where, for any k∈{1,2} and any $\left(r,s\right)\in {\left\{1,2,\;\ldots \;,m_{k}\right\}}^{2}$, $\tau_{rs}^{k}\equiv \tau \left(u_{k,r},u_{k,s}\right)$ is the so-called *encounter rate* of particles of $S_{k}$ in the states $u_{k,r}$ and $u_{k,s}$. It is the number of pairwise interactions between particles of $S_{k}$, that are in the states $u_{k,r}$ and $u_{k,s}$, per time unit, and for any sufficiently small Δt the product $\tau_{rs}^{k}\Delta t$ is the probability that one such interaction occur in the time interval Δt between individuals belonging to $S_{k}$ with states $u_{k,r}$ and $u_{k,s}$ respectively. As a consequence, $\tau_{rs}^{k}\Delta t$ is a *conditional* probability and $\tau_{rs}^{k}$ is the ratio of a conditional probability to time. Its presence in the first two terms at the right-hand side of each equation of system (Equation 1) is due to the fact that these terms express two basic properties:1.the increase in probability of state $u_{k,h}$ is the probability that some «candidate» particles of $S_{k}$ in the state $u_{k,r}$ interact with some «field» particles of $S_{k}$ in the state $u_{k,s}$: they fall in the state $u_{k,h}$ with a positive probability just in consequence of the interaction;2.the decrease in probability of state $u_{k,h}$ is the probability that some «test» particles of $S_{k}$ in the state $u_{k,h}$ interact with some «field» particles of $S_{k}$ in the state $u_{k,s}$; they leave the state $u_{k,h}$ with a positive probability only in consequence of the interaction.

The coefficients $\eta_{rs}^{\ell k}$ (for $r\in \{1,2,\;\ldots \;,m_{\ell }\}$ and $s\in \{1,2,\;\ldots \;,m_{k}\}$) in the other terms at the right-hand side of Equation (Equation 1) have the same meaning: for any $\left(\ell ,k\right)\in {\left\{1,2\right\}}^{2}$, $\eta_{rs}^{\ell k}$ is the *encounter rate* between any particle of $S_{\ell }$ which is in the state *r* and any particle of $S_{k}$ which is in the state *s*.

Notice that, as we shall do from now on to the end of the paper, the above system of equations has been identified by a single numerical label: from now on, each of the equations that compose a system will be distinguished by a subscript corresponding to its position in it: for instance, the third equation of system (X) will be indicated as ${\left(X\right)}_{3}$.

For the sake of completeness, it will be suitable to recall that it is customary in the literature to write system (Equation 1) in the form
(2)$$\frac{df_{h}^{\left(\ell \right)}}{dt}\left(t\right)=J_{h}^{\left(\ell \right)}\left[f^{\left(1\right)},f^{\left(2\right)}\right]\left(t\right)\;\left(h=1,2,\;\ldots \;,m_{\ell };\;\ell =1,2\right)\;,$$
where
(3)$$\begin{array}{cc} \; & J_{h}^{\left(\ell \right)}\left[f^{\left(1\right)},f^{\left(2\right)}\right]\left(t\right)=\\ \; & \;=G_{h}^{\left(\ell \right)}\left[f^{\left(1\right)},f^{\left(2\right)}\right]\left(t\right)-L_{h}^{\left(\ell \right)}\left[f^{\left(1\right)},f^{\left(2\right)}\right]\left(t\right)\;\left(\ell =1,2\right)\;, \end{array}$$
and
(4)$$\begin{array}{cc} \; & G_{h}^{\left(\ell \right)}\left[f^{\left(1\right)},f^{\left(2\right)}\right]\left(t\right)=\sum_{i,j}^{1\ldots m_{\ell }}\tau_{ij}^{\ell }F^{h}{}_{ij}\left(\ell \right)f_{i}^{\left(\ell \right)}\left(t\right)f_{j}^{\left(\ell \right)}\left(t\right)+\\ \; & \;+\sum_{i=1}^{m_{\ell }}\sum_{j=1}^{m_{k}}\eta_{ij}^{\ell k}\Phi^{h}{}_{ij}\left(\ell ,k\right)f_{i}^{\left(\ell \right)}\left(t\right)f_{j}^{\left(k\right)}\left(t\right)\;,\;\left(\ell =1,2\;;\;k\neq \ell \right)\\ \; & L_{h}^{\left(\ell \right)}\left[f^{\left(1\right)},f^{\left(2\right)}\right]\left(t\right)=f_{h}^{\left(\ell \right)}\left(t\right)\left[\sum_{i,j}^{1\ldots m_{\ell }}\tau_{hj}^{\ell }F_{hj}^{i}\left(\ell \right)f_{j}^{\left(\ell \right)}\left(t\right)+\right.\;,\\ \; & \;+\left.\sum_{i=1}^{m_{\ell }}\sum_{j=1}^{m_{k}}\eta_{hj}^{\ell k}\Phi_{hj}^{i}\left(\ell ,k\right)f_{j}^{\left(k\right)}\left(t\right)\right]\;\left(\ell =1,2\;;\;k\neq \ell \right) \end{array}$$

For obvious reasons, the term $G_{h}^{\left(\ell \right)}\left[f^{\left(1\right)},f^{\left(2\right)}\right]\left(t\right)$ is called *gain term*, while the term $L_{h}^{\left(\ell \right)}\left[f^{\left(1\right)},f^{\left(2\right)}\right]\left(t\right)$ is called *loss term*. Note that system (Equation 2) consists of $m=m_{1}+m_{2}$ equations.

## 3. Formulation and Interpretation of the Special Model

In this Section, the general model outlined in the previous one will be strongly modified under three aspects. A system *S* is considered, split into two subsystems $S_{1}$ and $S_{2}$, but

1.subsystem $S_{1}$, which from now on will be also referred to as «the system of subjects», consists of *only one* member, so that interactions between subjects are excluded as meaningless;2.subsystem $S_{2}$, which from now on will be also referred to as «the system of experiments», consists of *m* members (also called the «questions about the subject»), but again the interactions between these members are excluded;3.the state space of subjects will be denoted by $D\equiv \{u_{0},u_{1},\;\ldots \;u_{s}\}$;4.to *each* experiment in $S_{2}$ a *different* state space $D_{i}$ is associated, which is a set of positive integers (the outcomes of the experiment): we denote by $y_{i}$ the state variable in $D_{i}$ so that $D_{i}=\left\{y_{i1},y_{i2},\;\ldots \;,y_{im_{i}}\right\}$;5.interactions between the subject and experimental devices *do not* modify the states of the latter;6.the time variable is assumed to be *discrete*, i.e., the process described here is a *stepwise* process.

**Remark** **1.**
*In virtue of condition 1, $\tau_{ij}^{1}=0$ for any couple $\left(u_{i},u_{j}\right)$ of states in D.*


**Remark** **2.**
*It is necessary to point out, for the sake of the readability of the text, that some symbols introduced in Section 2 and appearing in system (Equation 1) as well as in definitions (Equation 4) must be modified because of condition 4. Precisely, the interaction rates $\tau_{hj}^{2}$ should be written $\tau_{h,k;j,r}^{2}$, where h and j are indexes identifying the state spaces $D_{h}$ and $D_{j}$ (hence the experiments) involved in the interaction, and k and r are the states (in $D_{h}$ and $D_{j}$, respectively) of interacting «questions». The same is true for the transition probabilities $F^{i}{}_{hj}\left(2\right)$ (to be written $F^{ri}{}_{r,h;s,j}\left(2\right)$). However, of course, in virtue of condition 2, this remark can be completely disregarded, since all the terms containing the considered interaction rates and transition probabilities vanish.*


**Remark** **3.**
*The situation is quite different for the «mixed» interaction rates $\eta_{ij}^{12}$ and $\eta_{ij}^{21}$ and transition probabilities $\Phi^{h}{}_{ij}\left(1,\;2\right)$ and $\Phi^{h}{}_{ij}\left(2,\;1\right)$, which should be written as $\eta_{i;kj}^{12}$, $\eta_{ki;j}^{21}$, $\Phi^{h}{}_{i;kj}\left(1,\;2\right)$ and $\Phi^{k;h}{}_{ki;j}\left(2,\;1\right)$, respectively, where k identifies in all cases the state space $D_{k}$ of the experiment involved in the interaction. Notice however that, in virtue of condition 5, $\Phi^{k;h}{}_{ki;j}\left(2,\;1\right)=\delta_{i}^{h}$ for any k∈{1,2,…,m} and any j∈{0,1,…,s}.*


**Remark** **4.**
*Analogously, $f_{h}^{\left(2\right)}$ must be written $f_{kh}^{\left(2\right)}$. This last symbol expresses the probability that the k-th experiment is in the h-th state $y_{kh}$ of $D_{k}$.*


**Remark** **5.**
*Finally, condition 6, according to which the evolution of the system will be described as a stepwise process allows us to consider a sequence ${\left\{t_{h}\right\}}_{h\in N}$ and to replace the time derivatives at the left-hand side of Equation (Equation 1) by finite differences of the form $f_{h}^{\left(1\right)}\left(t_{n+1}\right)-f_{h}^{\left(1\right)}\left(t_{n}\right)$ and $f_{k;h}^{\left(2\right)}\left(t_{n+1}\right)-f_{k;h}^{\left(2\right)}\left(t_{n}\right)$(n∈N), respectively. In particular, we shall take n=0.*


In view of the above remarks, system (Equation 1) simply becomes
(5)$$\begin{array}{cc} f_{h}^{\left(1\right)}\left(t_{1}\right)-f_{h}^{\left(1\right)}\left(t_{0}\right) & =\sum_{i=1}^{s}\sum_{k=1}^{m}[\sum_{j=1}^{m_{k}}\eta_{i;kj}^{12}\Phi^{h}{}_{i;kj}\left(1,2\right)f_{i}^{\left(1\right)}\left(t_{0}\right)f_{kj}^{\left(2\right)}\left(t_{0}\right)+\\ \; & -\eta_{h;kj}^{12}\Phi^{i}{}_{h;kj}\left(1,2\right)f_{h}^{\left(1\right)}\left(t_{0}\right)f_{kj}^{\left(2\right)}\left(t_{0}\right)]\;,\;\left(h=1,\;\ldots \;,s\right)\;,\\ f_{kh}^{\left(2\right)}\left(t_{1}\right)-f_{kh}^{\left(2\right)}\left(t_{0}\right) & =0\;,\;\;\;\;(k=1,\;\ldots \;,m\;;\;h=0,1,\;\ldots \;,m_{k})\;, \end{array}$$
to which the initial conditions
(6)$$\begin{array}{cc} \; & f_{h}^{\left(1\right)}\left(t_{0}\right)=f_{h0}^{\left(1\right)}\;\;\left(h=1,\;\ldots \;,s\right)\;.\\ \; & f_{kh}^{\left(2\right)}\left(t_{0}\right)=f_{kh0}^{\left(2\right)}\;\;\left(k=1,\;\ldots \;,m\;;\;h=0,1,\;\ldots \;,m_{k}\right) \end{array}$$
must be associated.

An interpretation as complete as possible of the special model obtained starting from conditions 1–6 will be outlined, not only in order to explain the meaning of these hypotheses, but also to turn the attention of the reader to the significant application of the system obtained above to the detection of depression. In this connection, what it will be necessary to stress is that system (Equation 5) is meant to describe the results of interactions between *information* rather than between objects. More precisely, the values $u_{i}$ of the variable in *D* should be interpreted as labels that *define*, in the appropriate context, possible states of the subject; then, the interaction between the subject and the *i*-th experience on it places this latter into a *state*, expressed by a real value $y_{ih}$, that is the outcome of a measurement of a physical parameter. Now, in virtue of the information carried by the value $y_{ih}$, the *i*-th experience *interacts* with the initial definition, in the sense that it can either confirm the initial state (by increasing its probability) or raise doubts about it (by lowering its probability). Accordingly, the terms $f_{i}^{\left(1\right)}\left(t_{0}\right)$ and $f_{kj}^{\left(2\right)}\left(t_{0}\right)$ in Equation (Equation 5) must be interpreted as the *absolute* probability that the *definition* of the state of the subject before the experience is the value $u_{i}$ and the *absolute* probability that the outcome of the *h*-th experience is the state $y_{kj}$, respectively; furthermore, the term $\Phi^{h}{}_{i;kj}\left(1,2\right)$ is the probability that definition $u_{i}$, after «interaction» with information $y_{kj}$
*becomes* definition $u_{h}$.

This outline of the methodological meaning of system (Equation 5) will be now explained in somehow more precise terms by referring to a special experimental problem connected to the detection of depression. In the next sections, by means of an example based on actual numerical data, it will be shown how the values of $f_{i}^{\left(1\right)}\left(t_{0}\right)$, $f_{kj}^{\left(2\right)}\left(t_{0}\right)$ and $\Phi^{h}{}_{i;kj}\left(1,2\right)$ should be assigned according to a suitable preliminary statistical analysis.

Therefore, the problem which gave rise to the present analysis will be now described, namely, the search for automated procedures to diagnose the possible depressive state of a human subject. As is well known to psychologists, depression is nowadays mainly diagnosed by means of questionnaires. Among them, of particular interest seems to be the one devised by Aaron Beck called BDI (Beck Depression Inventory). This is a list of 21 questions, each admitting four possible answers, with different scores varying from 0 to 3. According to the total score obtained by his/her answers, the subject is placed in a “class of depression”, namely, with a score from 1 to 10, the “normal class” (which will be denoted here as the *state 0*), with a score from 11 to 16, the “mild mood disturbance class” (the *state 1*), with a score from 17 to 20, the “borderline clinical depression class” (the *state 2*), with a score from 21 to 30, the “moderate depression class” (the *state 3*), with a score from 31 to 40, the “severe depression class” (the *state 4*), and finally, with a score over 40, the “extreme depression class” (the *state 5*).

Now, it is well known that any psychological state produces (and is expressed by) particular behaviors, which can be taken as “symptoms” of that state. These behaviors can be in turn described by the values of physical parameters, for instance voice fundamental frequencies (pitches) in a sequence of sufficiently small time intervals, or the number and the length of empty (or silent) pauses, and the speed of speech and the choice of words when reading a text or simply talking about everyday life (see [36], also for more complete references). Thus, the problem of establishing in which cases the measurements of these and other parameters confirm the results of BDI arises spontaneously, and in which cases they modify the conclusions drawn from the questionnaire. The final diagnosis should result from a suitable “superposition” of all these results. More precisely, one takes a subject initially placed by his (or her) BDI score in a state *k*, then tries to determine the value of one of the physical parameters that could reveal depression or a normal state: for instance, he submits him or her to a check of the pitches of his or her voice while reading a text, or of the number and the length of his or her empty pauses when telling his or her experiences during the last week. As laid out above, the parameters defined by these tests, that are random variables, will be denoted by the symbols $Y_{1}$, $Y_{2}$, …, $Y_{m}$. If the *i*-th test gives the outcome $Y_{i}=y_{ih}$ (i.e., the result of the *physical* interaction between the subject and the experimental device is $y_{ih}$), provided one has defined for any j∈D≡{0,1,…,5} the probability $\Phi^{j}{}_{k;ih}\left(1,2\right)\equiv \Phi^{j}{}_{kh}$ that this result “moves” the subject from the state *k* (initial diagnosis) to the state *j*, system (Equation 5) allows us to find a *final* probability distribution on *D* (in this connection, notice that the initial probability distribution vector ${\left(f_{h0}^{\left(1\right)}\right)}_{1\leq h\leq 5}\equiv {\left(f_{h0}\right)}_{1\leq h\leq 5}$ is assigned according to our confidence in the reliability of the BDI questionnaire: a good starting point could be to set $f_{k0}=1$ and $f_{h0}=0$ for any h≠k). Thus, the key step to make system (Equation 5) effective is to determine the transition probabilities $\Phi^{j}{}_{kh}\equiv \Phi^{j}{}_{k;ih}\left(1,2\right)$(j,k∈D). The way in which this can be performed will be discussed in detail in Section 4. At this moment, it will be appropriate to conclude the present Section with some important remarks.

First of all, it should be stressed that the first step to compute the transition probabilities $\Phi^{j}{}_{kh}\equiv \Phi^{j}{}_{k;ih}\left(1,2\right)$(j,k∈D) is to establish the range R(Y) of the values of *Y* that *confirm* the diagnosis delivered by BDI score, or, what is the same, to *synchronize* the responses of BDI and of the additional test. In other words, for any state *k*, one must know the class of values of the additional parameter *Y* that seems to “characterize”, in a suitable sense of the word, the state *k*. This will therefore be performed preliminarily in the next Section.

Thus, the procedure will enable the researchers to find the *final* probability distribution on the set of possible states of the subject, after *all* the tests devised to determine the level of his or her mental health.

**Remark** **6.**
*It should be carefully noted that the use of system (Equation 5) can be replaced by the repeated use of system*

(7)
$$\begin{array}{cc} f_{h}^{\left(1\right)}\left(t_{k}\right) & -f_{h}^{\left(1\right)}\left(t_{k-1}\right)=\sum_{i=1}^{s}[\sum_{j=1}^{m_{k}}\eta_{i;kj}^{12}\Phi^{h}{}_{i;kj}\left(1,2\right)f_{i}^{\left(1\right)}\left(t_{k-1}\right)f_{kj}^{\left(2\right)}\left(t_{k-1}\right)+\\ \; & -\eta_{h;kj}^{12}\Phi^{i}{}_{h;kj}\left(1,2\right)f_{h}^{\left(1\right)}\left(t_{k-1}\right)f_{kj}^{\left(2\right)}\left(t_{k-1}\right)]\;,\;\left(h=1,\;\ldots \;,s\right)\;,\\ f_{kh}^{\left(2\right)}\left(t_{k}\right) & -f_{kh}^{\left(2\right)}\left(t_{k-1}\right)=0\;,\;\;\;\;\left(h=0,1,\;\ldots \;,m_{k}\right)\;,\;\left(k=1,\;\ldots \;,m\right)\;. \end{array}$$

*More precisely, for any k∈{1,…,m}, the system for the parameter $Y_{k}$ can be written, where the values $f_{h}\left(t_{k-1}\right)$ are delivered by the same system applied to the random variable $Y_{k-1}$. By the way, this will be the procedure which will be adopted in Section 6 in our application of the method to the detection of depression in human subjects.*


**Remark** **7.***It should also be noted that the method outlined in Remark 6 and the use of system (Equation 5) are*  not *equivalent. The former is applied and illustrated in Section 6, while the latter* roughly *produces a «superposition» of the results obtained in the first step of the procedure shown therein.*

**Remark** **8.**
*Analogously, consider a sequence ${\left\{Y_{k}\right\}}_{1\leq k\leq m}$ of parameters. One cannot in general expect that, if Equation (Equation 7) is applied according to the plot $Y_{1}\to Y_{2}\to \;\ldots \;\to Y_{m}$ or according to a plot $Y_{i_{1}}\to Y_{i_{2}}\to \;\ldots \;Y_{i_{m}}$ (where $\left\{i_{1},i_{2},\;\ldots \;,i_{m}\right\}$ is any permutation of the set {1,2,…,m}), the same result will be obtained, i.e., the probability distributions ${\left\{f_{h}^{\left(1\right)}\left(t_{m}\right)\right\}}_{1\leq h\leq s}$ and ${\left\{f_{h}^{\left(1\right)}\left(t_{i_{m}}\right)\right\}}_{1\leq h\leq s}$ will be equal. As a matter of fact, as we shall see in Section 6, to obtain a single reliable distribution from the set of experiments, one needs to mediate between the different distributions obtained by following the different plots.*


## 4. Detecting Symptoms of Depression: General Formulation

In this Section, the model described in purely theoretical terms in the previous one will be applied to a set of data, in order to exhibit a concrete example of the results of the outlined method. Three physical (behavioral) parameters will be introduced: (a) the *total variation T* of pitch over the speech while reading a prescribed text; (b) the *average variationA* of pitch over the same reading, and (c) the percentage *O* of *inversions* —also called *oscillations*—of the sign of two subsequent differences of pitches over the whole reading), and system (Equation 7) will be applied to them.

Accordingly, first of all, the problem of «synchronizing» the values of these parameters with BDI labels must be solved. From a theoretical viewpoint, this must be performed by the following procedure. First, a sufficiently large sample *S* of both healthy and depressed subjects is considred. Setting n=|S| and denoting by *Y* any one of the parameters *T*, *A* and *O*, we also set

1.$R_{Y}=\left\{Y\left(x\right)\;|\;x\in S\right\}$ and $e^{\prime }\left(Y\right)=\min R\left(Y\right)$, $e^{\prime \prime }\left(Y\right)=\max R\left(Y\right)$;2.$C_{h}=\left\{x\in S\;|\;\mathrm{the}\;\mathrm{BDI}\;\mathrm{state}\;\mathrm{of}\;x\;\mathrm{is}\;h\right\}$;3.$|C_{h}|=n_{h}$ so that $n=\sum_{h=0}^{5}n_{h}$;4.$q_{h}\equiv n_{h}/n=P\left(C_{h}\right)$ the fraction of the subjects in *S* who are placed by BDI in the class $C_{h}$;5.$R_{h}\left(Y\right)=\left\{Y\left(x\right)\;|\;x\in C_{h}\right\}$ and $e_{h}^{\prime }\left(Y\right)=\min R_{h}\left(Y\right)$, $e_{h}^{\prime \prime }\left(Y\right)=\max R_{h}\left(Y\right)$;6.$\mu_{h}\left(Y\right)=\frac{1}{n_{h}}\sum_{x\in C_{h}}Y\left(x\right)$.7.$\sigma_{h}\left(Y\right)=\sqrt{\frac{1}{n_{h}}\sum_{x\in C_{h}}{\left[Y\left(x\right)-\mu_{h}\left(Y\right)\right]}^{2}}$.

In general, one expects that h≠k does not imply $\left[e_{h}^{\prime }\left(Y\right),e_{h}^{\prime \prime }\left(Y\right)\right]\cap \left[e_{k}^{\prime }\left(Y\right),e_{k}^{\prime \prime }\left(Y\right)\right]=⌀$. So, one has to determine, for any h∈{0,1,2,3,4,5}, a set $I_{h}\left(Y\right)$ in such a way that

1.for any *h*, $I_{h}\left(Y\right)\subseteq R_{h}\left(Y\right)$ is either a real interval or the join of a finite number of disjoint intervals;2.${\left\{I_{h}\left(Y\right)\right\}}_{0\leq h\leq 5}$ is a partition of $R_{Y}$;3.when a subject is assigned to class $C_{h}$ by BDI, then the corresponding value of *Y* has a high probability to be in $I_{h}\left(Y\right)$ (and a low probability to be in any interval $I_{k}\left(Y\right)$ for k≠h).

In the present context, the three conditions above are assumed to be satisfied, and any description of the procedure one should follow to meet them is omitted. The reader eager for details can usefully consult any university textbook about statistics and data analysis. In the next Section, however, an example of such a procedure will be shown in a simplified case.

Accordingly, a family of 6 sets $I_{h}\equiv I_{h}\left(Y\right)$(h=0,1,…,5), some of which will be possibly split in two intervals, is assumed to be given. We now denote by $n_{hk}$ the number of the subjects in $C_{h}$ for which the value of the parameter *Y* belongs to $I_{k}$, so that
$$n_{h}=\sum_{k=0}^{5}n_{hk}\;,$$
and, in accordance with condition 3,
$$n_{hh}>\sum_{k\neq h}n_{hk}\;.$$

Obviously,
$$p_{hk}\equiv P\left(I_{k}\;|\;C_{h}\right)=\frac{n_{hk}}{n_{h}}$$
and, setting $p_{h}\equiv p_{hh}$,
$$P\left(I_{h}^{c}\;|\;C_{h}\right)=\sum_{k\neq h}p_{hk}\left(Y\right)=1-p_{h}\;.$$

For the sake of completeness, it must be recalled that
(8)$$P\left(C_{h}\;|\;I_{k}\right)=\frac{p_{hk}q_{h}}{\sum_{i=0}^{5}p_{ik}q_{i}}=\frac{n_{hk}}{\sum_{i=0}^{5}n_{ik}}\;.$$

It must be stressed that (a) whenever an individual is taken randomly from the considered sample without checking its BDI class, given the value of *Y* associated to it, the above relation gives the probability that this individual belongs to any BDI class, and (b) the left-hand side of relation (Equation 8) only serves as a reminder of how one should compute the required probability from the sample data: if one decides to export the values of probabilities $q_{h}$ and $p_{hk}$ to the whole, potentially infinite population of past, present and future human beings, then the middle side of relation (Equation 8) must be used to express the probability $P(C_{h}\;|\;I_{k})$.

Now, for any k∈{0,…,5}, the law $P\left(\cdot \;|\;I_{k}\right)\;:\;h\in \left\{0,\;\ldots \;,5\right\}\;⟶\;P_{k}\left(h\right)=P\left(C_{h}\;|\;I_{k}\right)$ is a probability distribution on the “state space” {0,…,5} induced, for any examined subject, by the experimental result about the value of parameter *Y*, and it should be carefully noted that, when applied to the whole population, it *does not depend* on the BDI class to which the subject was assigned *a priori*, and this is quite natural since, as it has been just seen, each probability $P(C_{h}\;|\;I_{k})$ is computed under the assumption to ignore that class. In order to determine the probabilities $P(C_{h}\;|\;C_{r}\cap I_{k})$, that take into account the information about the BDI class to which the subject is assumed to belong before the experiment, one has to solve the following system of equations:$$P\left(C_{h}\;|\;I_{k}\right)=\sum_{r=0}^{5}P\left(C_{h}\;|\;C_{r}\cap I_{k}\right)P\left(C_{r}\;|\;I_{k}\right)\;,$$
that is
(9)$$n_{hk}=\sum_{r=0}^{5}n_{rk}P\left(C_{h}\;|\;C_{r}\cap I_{k}\right)\;.$$

Notice that, referring to system (Equation 7), one has $P\left(C_{h}\;|\;C_{r}\cap I_{j}\right)=\Phi^{h}{}_{r;j}$ (where the index relative to the parameter has been dropped as useless for the sake of simplicity), so that the above system gives the transition probabilities needed to use system (Equation 7).

## 5. Detecting Symptoms of Depression: Computing Parameters on a Sample Study

Now, system (Equation 9) can be easily solved if one has enough time and patience, or simply a computer. Nevertheless, the general case will be left aside and the above procedure will be now applied to a reference sample already used to the same aim in [34]. It could be objected that the considered sample cannot be defined “sufficiently large”. As a matter of fact, the present application aims to be nothing more than an example of the way in which the method should work, and a suitable extension (and better validation) of the method outlined here is planned to be the object of a forthcoming paper. Furthermore, just in connection with the small size of our sample, the consideration of all the five possible states defined by BDI will be renounced, and only *two* states will be considered, 0= healthy, and 1= depressed. This simplifies the procedure and, at the same time, can provide the reader with a sufficiently clear plot of the method.

So, in the case under consideration, we have n≡|S|=22, $n_{0}\equiv \;|C_{0}|\;=n_{1}\equiv \left|C_{1}\right|=11$. Consider first the parameter *T*, for which $I_{0}\left(T\right)=\left[285.79,422.88\right]$, $I_{1}\left(T\right)=\left(115.88,285.79\right]\cup \left[422.88,1005.01\right)$, and
(10)$$\begin{array}{cc} n_{00}\left(T\right) & =7\\ n_{01}\left(T\right) & =4\\ n_{10}\left(T\right) & =2\\ n_{11}\left(T\right) & =9\;. \end{array}$$

System (Equation 9) takes the form
(11)$$\begin{array}{cc} n_{00} & =n_{00}P\left(C_{0}\;|\;C_{0}\cap I_{0}\right)+n_{10}P\left(C_{0}\;|\;C_{1}\cap I_{0}\right)\\ n_{01} & =n_{01}P\left(C_{0}\;|\;C_{0}\cap I_{1}\right)+n_{11}P\left(C_{0}\;|\;C_{1}\cap I_{1}\right)\\ n_{10} & =n_{00}P\left(C_{1}\;|\;C_{0}\cap I_{0}\right)+n_{10}P\left(C_{1}\;|\;C_{1}\cap I_{0}\right)\\ n_{11} & =n_{01}P\left(C_{1}\;|\;C_{0}\cap I_{1}\right)+n_{11}P\left(C_{1}\;|\;C_{1}\cap I_{1}\right)\;. \end{array}$$
that is
(12)$$\begin{array}{cc} 7 & =7P\left(C_{0}\;|\;C_{0}\cap I_{0}\right)+2P\left(C_{0}\;|\;C_{1}\cap I_{0}\right)\\ 4 & =4P\left(C_{0}\;|\;C_{0}\cap I_{1}\right)+9P\left(C_{0}\;|\;C_{1}\cap I_{1}\right) \end{array}\;,$$
because a straightforward calculation, based on the relation $P\left(C_{1}\;|\;C_{0}\cap I_{0}\right)=1-P\left(C_{0}\;|\;C_{0}\cap I_{0}\right)$, shows at once that system (Equation 11) is redundant, and that the third and the fourth of Equation (Equation 11) are reproductions of the first and second ones respectively.

Now, as a system of two equations with four unknowns, system (Equation 12) does not allow us to determine all the required transition probabilities: two of them must be assigned as parameters. Therefore, setting $P(C_{0}\;|\;C_{0}\cap I_{0})=\lambda$ and $P(C_{1}\;|\;C_{1}\cap I_{1})=\mu$, system (Equation 11) becomes
(13)$$\begin{array}{cc} P(C_{0}\;|\;C_{1}\cap I_{0}) & =\frac{7}{2}\left(1-\lambda \right)\\ P(C_{0}\;|\;C_{0}\cap I_{1}) & =\frac{9\mu -5}{4} \end{array}\;,$$
and the parameters must satisfy the conditions
(14)$$\begin{array}{cc} \; & \frac{5}{7}\leq \lambda \leq 1\;,\\ \; & \frac{5}{9}\leq \mu \leq 1\;. \end{array}$$

The above procedure can be reproduced step by step for each of the two remaining parameters *A* and *O*. One only needs to insert in system (Equation 11) the values of $n_{hk}$ resulting from Tables 1–6 in [34] for these parameters. These values are listed in the following Table 1 (while, for the sake of completeness, the intervals $I_{0}$ and $I_{1}$ are specified for all the three parameters *T*, *A* and *O* in Table 2, Table 3 and Table 4).

So, with the same definition of parameters λ and μ and the same calculations as before, we find for λ and μ the conditions collected in Table 5.

**Remark** **9.**
*As probabilities, λ and μ belong to [0,1], and the above conditions, as derived by means of straightforward algebraic computations, simply give lower bounds to their values. As a consequence, the negative values in the third column simply mean that, for the parameter O, λ and μ are free to run over the whole interval [0,1].*


Finally, it is possible to write the matrices of the transition probabilities for all three parameters in consideration. More precisely,
(15)$$\Phi_{0}\equiv \left(\Phi^{r}{}_{s0}\right)=\left(\begin{array}{cc} \lambda & 1-\lambda \\ \frac{7}{2}\left(1-\lambda \right) & \frac{7}{2}\lambda -\frac{5}{2} \end{array}\right)\;,\;\Phi_{1}\equiv \left(\Phi^{r}{}_{s1}\right)=\left(\begin{array}{cc} \frac{9\mu -5}{4} & \frac{9}{4}\left(1-\mu \right)\\ 1-\mu & \mu \end{array}\right)\;,\;$$
for the parameter *T*;
(16)$$\Phi_{0}\equiv \left(\Phi^{r}{}_{s0}\right)=\left(\begin{array}{cc} \lambda & 1-\lambda \\ 4(1-\lambda ) & 4\lambda -3 \end{array}\right)\;,\;\Phi_{1}\equiv \left(\Phi^{r}{}_{s1}\right)=\left(\begin{array}{cc} 3\mu -2 & 3(1-\mu )\\ 1-\mu & \mu \end{array}\right)\;,\;$$
for the parameter *A*, and
(17)$$\Phi_{0}\equiv \left(\Phi^{r}{}_{s0}\right)=\left(\begin{array}{cc} \lambda & 1-\lambda \\ \frac{7}{8}\left(1-\lambda \right) & \frac{1}{8}\left(1+7\lambda \right) \end{array}\right)\;,\;\Phi_{1}\equiv \left(\Phi^{r}{}_{s1}\right)=\left(\begin{array}{cc} \frac{3}{4}\left(1-\mu \right) & \frac{1}{4}\left(1+3\mu \right)\\ 1-\mu & \mu \end{array}\right)\;,\;$$
for the parameter *O*.

**Remark** **10.**
*Notice that, though the same symbols λ and μ have been used for all three parameters T, A and O, they need not to take the same values in all cases. On the contrary, λ and μ will take in general different values from case to case. The reason for the differences will be explained in Remark 11.*


**Remark** **11.**
*The meaning of the parametric nature of the solution as well as of these last conditions now requires careful discussion. As a matter of fact, beyond the mere algebraic condition, there is at least one deep conceptual reason why two of the above transition probabilities cannot be determined only on the base of the experienced relative frequencies of intervals $I_{k}$: a careful check of the above conditions on parameters, as well as of the consequences of assigning to them either the least or the greatest value allowed, shows that the choice of the values for λ and μ is strictly related to our trust in the effectiveness of our experiments. More precisely, consider for simplicity only parameter T, and assume to have chosen λ=5/7. It follows that $P(C_{0}\;|\;C_{0}\cap I_{0})=5/7$, $P(C_{1}\;|\;C_{0}\cap I_{0})=3/7$, $P(C_{0}\;|\;C_{1}\cap I_{0})=1$ and $P(C_{1}\;|\;C_{1}\cap I_{0})=0$. In addition, set μ=1. It follows that $P(C_{0}\;|\;C_{0}\cap I_{1})=1$, $P(C_{1}\;|\;C_{0}\cap I_{1})=0$, $P(C_{0}\;|\;C_{1}\cap I_{1})=0$ and $P(C_{1}\;|\;C_{1}\cap I_{1})=1$. So, this choice for parameters λ and μ corresponds to ascribing to the experiment an heavy bias towards the result «healthy» (or «not depressed»). Indeed, a subject classified as «healthy» by BDI is still classified «healthy» after the experiment even if the value of T belongs to $I_{1}$, while a subject classified as «depressed» by BDI is classified «healthy» after the experiment when the value of T belongs to $I_{0}$. Conversely, choose λ=1 and μ=5/9. One finds $P(C_{0}\;|\;C_{0}\cap I_{0})=1$, $P(C_{1}\;|\;C_{0}\cap I_{0})=0$, $P(C_{0}\;|\;C_{1}\cap I_{0})=0$, $P(C_{1}\;|\;C_{1}\cap I_{0})=1$, $P(C_{0}\;|\;C_{0}\cap I_{1})=0$, $P(C_{1}\;|\;C_{0}\cap I_{1})=1$, $P(C_{0}\;|\;C_{1}\cap I_{1})=4/9$ and $P(C_{1}\;|\;C_{1}\cap I_{1})=5/9$, which clearly correspond to a bias towards the state «depressed».*

*These considerations can be repeated word by word for parameters A and O, and lead us to the conclusion that in each case the values of parameters λ and μ must be «tuned» according to our trust in the effectiveness of our experiment. At a first glance, the choice that seems not to introduce biases is*

$$\begin{array}{cc} & \lambda^{*}=\frac{\lambda_{\min}+\lambda_{\max}}{2}\\ & \mu^{*}=\frac{\mu_{\min}+\mu_{\max}}{2}\;. \end{array}$$


*Of course, this choice requires to be revised and corrected after a sufficiently large number of additional experiments, by comparison with other evidence.*


**Remark** **12.**
*It is readily seen that with the above choice of the values of λ and μ the matrices (Equation 15)–(Equation 17) assume the forms*

(18)
$$\Phi_{0}\equiv \left(\Phi^{r}{}_{s0}\right)=\left(\begin{array}{cc} 6/7 & 1/7\\ 1/2 & 1/2 \end{array}\right)\;,\;\Phi_{1}\equiv \left(\Phi^{r}{}_{s1}\right)=\left(\begin{array}{cc} 1/2 & 1/2\\ 2/9 & 7/9 \end{array}\right)\;,$$

*for the parameter T;*

(19)
$$\Phi_{0}\equiv \left(\Phi^{r}{}_{s0}\right)=\left(\begin{array}{cc} 7/8 & 1/8\\ 1/2 & 1/2 \end{array}\right)\;,\;\Phi_{1}\equiv \left(\Phi^{r}{}_{s1}\right)=\left(\begin{array}{cc} 1/2 & 1/2\\ 1/6 & 5/6 \end{array}\right)\;,$$

*for the parameter A, and*

(20)
$$\Phi_{0}\equiv \left(\Phi^{r}{}_{s0}\right)=\left(\begin{array}{cc} 1/2 & 1/2\\ 7/16 & 9/16 \end{array}\right)\;,\;\Phi_{1}\equiv \left(\Phi^{r}{}_{s1}\right)=\left(\begin{array}{cc} 3/8 & 5/8\\ 1/2 & 1/2 \end{array}\right)\;,$$

*for the parameter O, respectively.*

*Now, it is to be noted that while the transition matrices for parameters T and A have one row (1,0) or (0,1) corresponding to a certain transition of the subject from the state $C_{1}$ to the state $C_{0}$ or vice versa according to whether the value of the parameter belongs to $I_{0}$ or to $I_{1}$, the corresponding rows for the parameter O are in both cases (1/2,1/2), expressing a complete uncertainty about the «final» state of the examined subject. This allows us to state that parameters T and A contribute information much richer than that transmitted by O, so that the computation of the value of parameter O could also be given up, or should be used to «put in doubt» the results of any previous sequence of experiments. This point will be illustrated more clearly through examples in the next Section.*


## 6. Detecting Symptoms of Depression: How to Classify New Cases

The present Section is devoted to show the way in which the above model and the data obtained from the basic sample can be used to update the probability distribution on the states 0 (healthy) and 1 (depressed) for any additional case. In particular, two subjects not contained in the starting sample will be considered here, the former assigned by BDI in the class $C_{1}$ (depressed), the latter assigned instead to the class $C_{0}$ (healthy).

The first case, who from now on will be referred to as “S37”, a forty year-old female labeled without uncertainty as «depressed» by BDI, has obtained for the parameters *T*, *A* and *O* the values shown in Table 6, which shows that S37 is depressed according to each experiment.

Using these results, apply system (Equation 7) to each parameter for subject S37. As laid out in Section 4, we choose to repeatedly apply system (Equation 7), first to *T*, next to *A* and finally to *O*. Starting with parameter *T*, one has
(21)$$\begin{array}{cc} f_{0}^{\left(1\right)}\left(t_{1}\right) & =\Phi^{0}{}_{1;1}\left(1,2\right)f_{1}^{\left(1\right)}\left(t_{0}\right)f_{1}^{\left(2\right)}\left(t_{1}\right)=\frac{2}{9}\approx 22\%\\ f_{1}^{\left(1\right)}\left(t_{1}\right) & =1-\Phi^{0}{}_{1;1}\left(1,2\right)f_{1}^{\left(1\right)}\left(t_{0}\right)f_{1}^{\left(2\right)}\left(t_{1}\right)=\frac{7}{9}\approx 78\%\;. \end{array}$$

By applying the same system to *A*, starting from the updated probability distribution, we obtain
(22)$$\begin{array}{cc} f_{0}^{\left(1\right)}\left(t_{1}\right) & =\frac{2}{9}+\Phi^{0}{}_{1;1}\left(1,2\right)f_{1}^{\left(1\right)}\left(t_{0}\right)f_{1}^{\left(2\right)}\left(t_{1}\right)-\Phi^{1}{}_{0;1}\left(1,2\right)f_{0}^{\left(1\right)}\left(t_{0}\right)f_{1}^{\left(2\right)}\left(t_{1}\right)=\\ \; & =\frac{2}{9}+\frac{1}{6}\cdot \frac{7}{9}-\frac{1}{2}\cdot \frac{2}{9}=\frac{13}{54}\approx 24\%\\ f_{1}^{\left(1\right)}\left(t_{1}\right) & =\frac{41}{54}\approx 76\%\;. \end{array}$$

Again, by repeating step-by-step the above procedure for the parameter *O*, we find
(23)$$\begin{array}{cc} f_{0}^{\left(1\right)}\left(t_{1}\right) & =\frac{13}{54}+\Phi^{0}{}_{1;1}\left(1,2\right)f_{1}^{\left(1\right)}\left(t_{0}\right)f_{1}^{\left(2\right)}\left(t_{1}\right)-\Phi^{1}{}_{0;1}\left(1,2\right)f_{0}^{\left(1\right)}\left(t_{0}\right)f_{1}^{\left(2\right)}\left(t_{1}\right)=\\ \; & =\frac{13}{54}+\frac{2}{3}\cdot \frac{41}{54}-\frac{1}{2}\cdot \frac{13}{54}=\frac{203}{324}\approx 63\%\\ f_{1}^{\left(1\right)}\left(t_{1}\right) & =\frac{255}{432}\approx 37\%\;. \end{array}$$

In conclusion, the probability distribution on the «sample space» {0,1} can be updated according the following plot, where in any pair of values, the first is the probability of the healthy state, while the second is that of the depressed state:$$⟼\;\;\;\;\;\;\;\;\;\;\;{\;}^{BDI}\;\;\;\;\;\;\left(0,1\right)\;⟶\;\;\;\;\;\;\;\;{\;}^{T}\;\;\;\;\left(0.22,0.78\right)\;⟶\;\;\;\;\;\;\;\;{\;}^{A}\;\;\;\;\left(0.24,0.76\right)\;⟶\;\;\;\;\;\;\;\;{\;}^{O}\;\;\;\;\left(0.47,0.53\right)$$
$$⟼\;\;\;\;\;\;\;\;\;\;\;{\;}^{BDI}\;\;\;\;\;\;\left(0,1\right)\;⟶\;\;\;\;\;\;\;\;{\;}^{T}\;\;\;\;\left(0.22,0.78\right)\;⟶\;\;\;\;\;\;\;\;{\;}^{O}\;\;\;\;\left(0.47,0.53\right)\;⟶\;\;\;\;\;\;\;\;{\;}^{A}\;\;\;\;\left(0.32,0.68\right)$$
$$⟼\;\;\;\;\;\;\;\;\;\;\;{\;}^{BDI}\;\;\;\;\;\;\left(0,1\right)\;⟶\;\;\;\;\;\;\;\;{\;}^{A}\;\;\;\;\left(0.17,0.83\right)\;⟶\;\;\;\;\;\;\;\;{\;}^{T}\;\;\;\;\left(0.27,0.73\right)\;⟶\;\;\;\;\;\;\;\;{\;}^{O}\;\;\;\;\left(0.42,0.58\right)$$
$$⟼\;\;\;\;\;\;\;\;\;\;\;{\;}^{BDI}\;\;\;\;\;\;\left(0,1\right)\;⟶\;\;\;\;\;\;\;\;{\;}^{A}\;\;\;\;\left(0.17,0.83\right)\;⟶\;\;\;\;\;\;\;\;{\;}^{O}\;\;\;\;\left(0.48,0.52\right)\;⟶\;\;\;\;\;\;\;\;{\;}^{T}\;\;\;\;\left(0.36,0.64\right)$$
$$⟼\;\;\;\;\;\;\;\;\;\;\;{\;}^{BDI}\;\;\;\;\;\;\left(0,1\right)\;⟶\;\;\;\;\;\;\;\;{\;}^{O}\;\;\;\;\left(0.50,0.50\right)\;⟶\;\;\;\;\;\;\;\;{\;}^{T}\;\;\;\;\left(0.36,0.64\right)\;⟶\;\;\;\;\;\;\;\;{\;}^{A}\;\;\;\;\left(0.29,0.71\right)$$
$$⟼\;\;\;\;\;\;\;\;\;\;\;{\;}^{BDI}\;\;\;\;\;\;\left(0,1\right)\;⟶\;\;\;\;\;\;\;\;{\;}^{O}\;\;\;\;\left(0.50,0.50\right)\;⟶\;\;\;\;\;\;\;\;{\;}^{A}\;\;\;\;\left(0.33,0.67\right)\;⟶\;\;\;\;\;\;\;\;{\;}^{T}\;\;\;\;\left(0.31,0.69\right).$$

These paths from initial health and depression probabilities, as assigned to subject S37 by BDI, to final probabilities obtained according to different test arrangements are illustrated in Figure 2.

The second subject examined here, who from now on will be referred to as “S08”, is labelled (without uncertainty) as healthy by BDI, but turns out to be depressed by all the parameters *T*, *A* and *O*, as results from Table 7.

For subject S08, the following list of plots is obtained:$$⟼\;\;\;\;\;\;\;\;\;\;\;{\;}^{BDI}\;\;\;\;\;\;\left(1,0\right)\;⟶\;\;\;\;\;\;\;\;{\;}^{T}\;\;\;\;\left(0.50,0.50\right)\;⟶\;\;\;\;\;\;\;\;{\;}^{A}\;\;\;\;\left(0.33,0.67\right)\;⟶\;\;\;\;\;\;\;\;{\;}^{O}\;\;\;\;\left(0.46,0.54\right)$$
$$⟼\;\;\;\;\;\;\;\;\;\;\;{\;}^{BDI}\;\;\;\;\;\;\left(1,0\right)\;⟶\;\;\;\;\;\;\;\;{\;}^{T}\;\;\;\;\left(0.50,0.50\right)\;⟶\;\;\;\;\;\;\;\;{\;}^{O}\;\;\;\;\left(0.44,0.56\right)\;⟶\;\;\;\;\;\;\;\;{\;}^{A}\;\;\;\;\left(0.31,0.69\right)$$
$$⟼\;\;\;\;\;\;\;\;\;\;\;{\;}^{BDI}\;\;\;\;\;\;\left(1,0\right)\;⟶\;\;\;\;\;\;\;\;{\;}^{A}\;\;\;\;\left(0.50,0.50\right)\;⟶\;\;\;\;\;\;\;\;{\;}^{T}\;\;\;\;\left(0.36,0.64\right)\;⟶\;\;\;\;\;\;\;\;{\;}^{O}\;\;\;\;\left(0.47,0.53\right)$$
$$⟼\;\;\;\;\;\;\;\;\;\;\;{\;}^{BDI}\;\;\;\;\;\;\left(1,0\right);⟶\;\;\;\;\;\;\;\;{\;}^{A}\;\;\;\;\left(0.50,0.50\right)\;⟶\;\;\;\;\;\;\;\;{\;}^{O}\;\;\;\;\left(0.44,0.56\right)\;⟶\;\;\;\;\;\;\;\;{\;}^{T}\;\;\;\;\left(0.34,0.66\right)$$
$$⟼\;\;\;\;\;\;\;\;\;\;\;{\;}^{BDI}\;\;\;\;\;\;\left(1,0\right)\;⟶\;\;\;\;\;\;\;\;{\;}^{O}\;\;\;\;\left(0.38,0.62\right)\;⟶\;\;\;\;\;\;\;\;{\;}^{T}\;\;\;\;\left(0.33,0.67\right)\;⟶\;\;\;\;\;\;\;\;{\;}^{A}\;\;\;\;\left(0.28,0.72\right)$$
$$⟼\;\;\;\;\;\;\;\;\;\;\;{\;}^{BDI}\;\;\;\;\;\;\left(1,0\right)\;⟶\;\;\;\;\;\;\;\;{\;}^{O}\;\;\;\;\left(0.38,0.62\right)\;⟶\;\;\;\;\;\;\;\;{\;}^{A}\;\;\;\;\left(0.29,0.71\right)\;⟶\;\;\;\;\;\;\;\;{\;}^{T}\;\;\;\;\left(0.30,0.70\right).$$

These paths from initial health and depression probabilities, as assigned to subject S08 by BDI, to final probabilities obtained according to different test arrangements are illustrated in Figure 3.

Finally, consider again a subject—who will be identified as S15—labelled as depressed by BDI, but described as healthy by two indicators and as depressed by only one of them according to Table 8 below.

For subject S15, the following list of plots follows:$$⟼\;\;\;\;\;\;\;\;\;\;\;{\;}^{BDI}\;\;\;\;\;\;\left(0,1\right)\;⟶\;\;\;\;\;\;\;\;{\;}^{T}\;\;\;\;\left(0.50,0.50\right)\;⟶\;\;\;\;\;\;\;\;{\;}^{A}\;\;\;\;\left(0.33,0.67\right)\;⟶\;\;\;\;\;\;\;\;{\;}^{O}\;\;\;\;\left(0.46,0.54\right)$$
$$⟼\;\;\;\;\;\;\;\;\;\;\;{\;}^{BDI}\;\;\;\;\;\;\left(0,1\right)\;⟶\;\;\;\;\;\;\;\;{\;}^{T}\;\;\;\;\left(0.50,0.50\right)\;⟶\;\;\;\;\;\;\;\;{\;}^{O}\;\;\;\;\left(0.47,0.53\right)\;⟶\;\;\;\;\;\;\;\;{\;}^{A}\;\;\;\;\left(0.32,0.68\right)$$
$$⟼\;\;\;\;\;\;\;\;\;\;\;{\;}^{BDI}\;\;\;\;\;\;\left(0,1\right)\;⟶\;\;\;\;\;\;\;\;{\;}^{A}\;\;\;\;\left(0.17,0.83\right)\;⟶\;\;\;\;\;\;\;\;{\;}^{T}\;\;\;\;\left(0.56,0.44\right)\;⟶\;\;\;\;\;\;\;\;{\;}^{O}\;\;\;\;\left(0.47,0.53\right)$$
$$⟼\;\;\;\;\;\;\;\;\;\;\;{\;}^{BDI}\;\;\;\;\;\;\left(0,1\right);⟶\;\;\;\;\;\;\;\;{\;}^{A}\;\;\;\;\left(0.17,0.83\right)\;⟶\;\;\;\;\;\;\;\;{\;}^{O}\;\;\;\;\left(0.45,0.55\right)\;⟶\;\;\;\;\;\;\;\;{\;}^{T}\;\;\;\;\left(0.66,0.34\right)$$
$$⟼\;\;\;\;\;\;\;\;\;\;\;{\;}^{BDI}\;\;\;\;\;\;\left(0,1\right)\;⟶\;\;\;\;\;\;\;\;{\;}^{O}\;\;\;\;\left(0.44,0.56\right)\;⟶\;\;\;\;\;\;\;\;{\;}^{T}\;\;\;\;\left(0.66,0.34\right)\;⟶\;\;\;\;\;\;\;\;{\;}^{A}\;\;\;\;\left(0.39,0.61\right)$$
$$⟼\;\;\;\;\;\;\;\;\;\;\;{\;}^{BDI}\;\;\;\;\;\;\left(0,1\right)\;⟶\;\;\;\;\;\;\;\;{\;}^{O}\;\;\;\;\left(0.44,0.56\right)\;⟶\;\;\;\;\;\;\;\;{\;}^{A}\;\;\;\;\left(0.31,0.69\right)\;⟶\;\;\;\;\;\;\;\;{\;}^{T}\;\;\;\;\left(0.61,0.39\right).$$

These paths from initial health and depression probabilities, as assigned to subject S15 by BDI, to final probabilities obtained according to different test arrangements are illustrated in Figure 4.

According to the above results, it seems that the probabilities of health and depression for each subject, when updated according to the outcomes of tests about indicators *T*, *A* and *O*, depend on the order in which the values of indicators are registered and compared. Nevertheless, at least when the indicators are all in accordance, the variations in the final probability distributions are very limited. This suggests referring to the average probabilities (on the set of all possible permutations of the triple (T,A,O)) rather than choosing any one of the values listed in the final column, and to compute explicitly the standard deviation in order to get a quantitative expression of their spreading. More precisely, let us denote by $\mu_{h}$ and $\mu_{d}$, respectively, the average values of probabilities for healthy and depressed state at each step (**I**, **II** or **III**), and by σ all the corresponding standard deviations. Thus, postponing to the next Section a discussion of the meaning of the results, we may collect them in Table 9.

The results on $\mu_{h}$ (and, consequently, on $\mu_{d}$) listed in the above table are illustrated in Figure 5.

## 7. Discussion and Research Perspectives

As it has been seen in Section 6, the order in which the values of parameters *T*, *A* and *O* are considered influences the resulting probability distribution on the set {healthy,depressed}. Nevertheless, when the values of all these parameters are in agreement (i.e., they assign a subject to the same BDI class), the procedure turns out to be approximately «commutative», in the sense that the probability distributions obtained by following the different plots differ very little from each other. A perceivable difference can be observed when some parameters confirm BDI classification and others refute it. It is precisely this circumstance that suggests to use the averages of the outcomes of different plots, taking also into account the very small values of standard deviations at each step.

At a first glance, it can be considered as a loss of information to have transformed the BDI classification, which is assumed to be sharp, into a probabilistic statement. In this connection, however, it must be recalled that, strictly speaking, BDI classification is in turn probabilistic, and our choice to assign the values 0 and 1 to the initial probabilities simply expresses our absolute confidence in the response of the questionnaire. If one questions at least partially the veracity of this decision, and make different hypotheses about the reliability of the additional indicators (for instance, assuming them more accurate in negative evaluations than in positive ones or vice versa), so that parameters λ and μ are given different values, then the probabilistic character of all the results becomes evident.

Naturally, the assignment of the values of the initial probabilities and of the parameters λ and μ should not be thought of as definitive, but in turn subjected to repeated updates by further experiments on more subjects. This immediately defines the natural development of the research and the problems to be tackled.

First, it is evident that the reference sample used to define the confidence intervals of the parameters introduced here should be greatly expanded; a sufficient expansion would certainly modify the ranges of possible values for λ and μ and their choice. It can be expected that after a sufficiently large number of examined subjects, both ranges and choices will stabilize, especially if one establishes the «relative weight» that should be assigned to the BDI initial classification and to the values of parameters *T*, *A* and *O*, respectively. So, the first step in the future development of this research must be to increase the size of the reference sample. In addition, the values of λ and μ should be arbitrarily changed in order to determine through experimental controls the different degrees of belief that could be assigned to BDI and to parameters *T*, *A* and *O*, respectively.

In connection with the expansion of the sample, of particular interest would be to conduct an analysis of the depressive effects of COVID on a large sample, both through the BDI and through speech analysis. Unfortunately, we do not yet have sufficient data on this subject, but we intend to develop research in this direction as soon as we have collected enough data.

A natural second step will then consist in automatizing the computation of the transition matrices and, as a consequence, of the solutions of Equation (Equation 7)${}_{1}$. In such a way, Equation (Equation 7)${}_{1}$ will behave similarly to a tool for training (on the reference sample) and then to use on any subsequent subject in the form of software for diagnosing depression.

Finally, the question about the possibility of reproducing the method presented here for any kind of diagnostic problems arises spontaneously, once the experimental devices and the parameters of interest will be defined for them. The answer can be readily seen to be affirmative. A very good example of this kind of development of the studies about possible applications of the method presented here can be found in [37,38], where a study of mental health based on an electroencephalogram headset is proposed: the application of such a study should be of the greatest interest with regard to depression and the comparison of its results with those of the BDI using the method presented here. A more general exposition of the method, showing at once its independence of the context, and its effectiveness in solving any classification problem, is given in [14].

## Figures and Tables

**Figure 1 brainsci-13-01339-f001:**
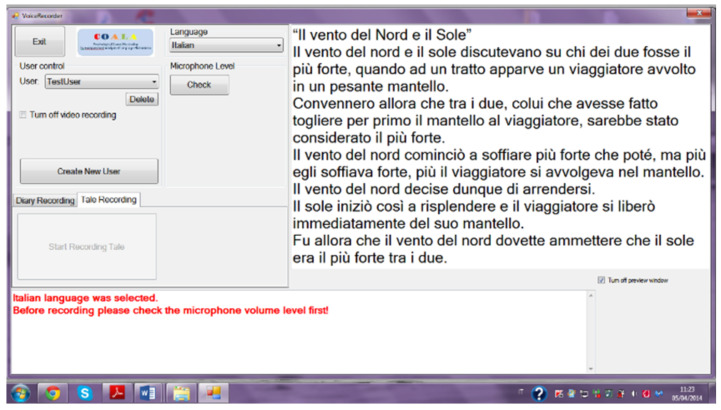
The Aesop’s tale “The North Wind and the Sun” as it was presented to the participants involved in the data collection.

**Figure 2 brainsci-13-01339-f002:**
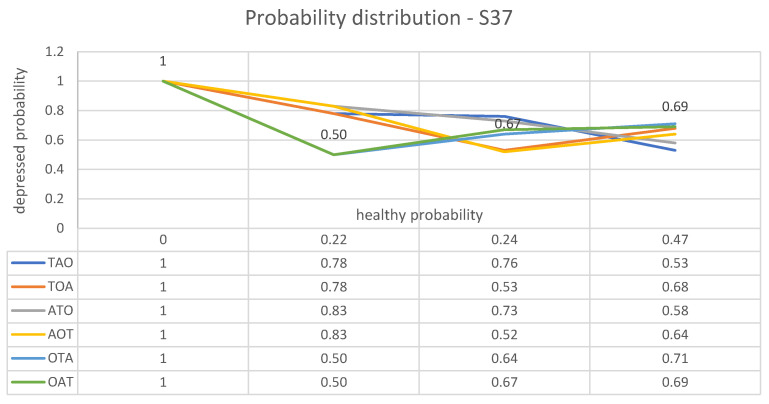
Graphic representation of the way in which the estimated probabilities of health and depression for subject S37 vary after each speech test, depending on the different relative importance assigned to each measured parameter. The coordinates of the edges of each plot are the health and depression probabilities of subject S37 after each test. More precisely, the *x*-coordinate is the health probability and the *y*-coordinate is the depression probability.

**Figure 3 brainsci-13-01339-f003:**
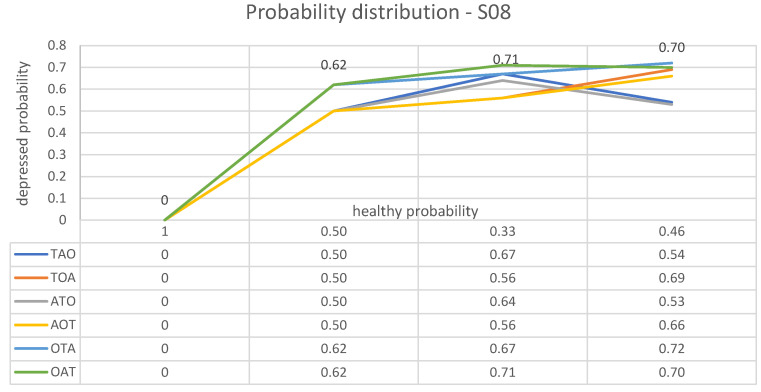
Graphic representation of the way in which the estimated probabilities of health and depression for subject S08 vary after each speech test, depending on the different relative importance assigned to each measured parameter.

**Figure 4 brainsci-13-01339-f004:**
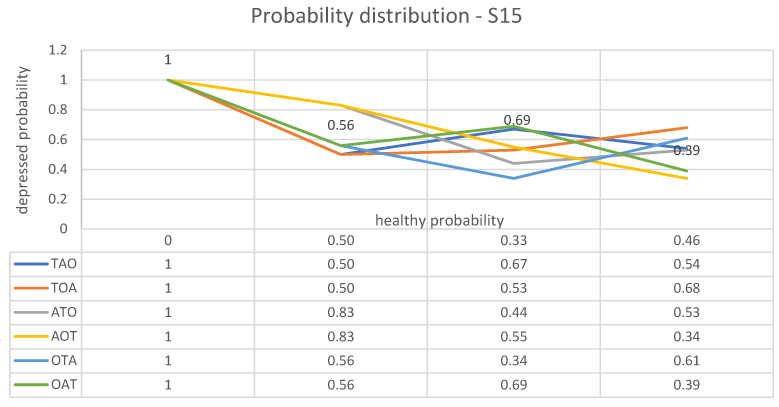
Graphic representation of the way in which the estimated probabilities of health and depression for subject S15 vary after each speech test, depending on the different relative importance assigned to each measured parameter.

**Figure 5 brainsci-13-01339-f005:**
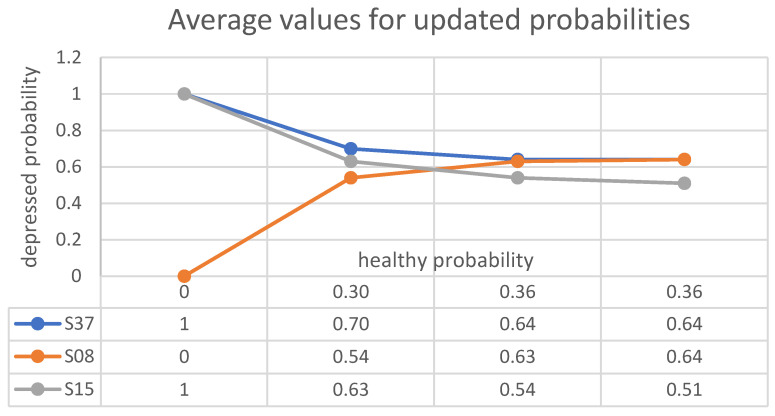
The plots above depict the evolution of the *average* depression probabilities for subjects S37, S08 and S15. The coordinates of the edges of the plot are the average health and depression probabilities of these subjects at each step, as updated by the *triples* of outcomes of experiments. A comparison of this picture with Figure 2, Figure 3 and Figure 4 highlights that the final results of the different test arrangements are much more dispersed around the average results for subjects S08 and S15 than for subject S37. This seems to reflect the presence at each step of results in apparent mutual contrast.

**Table 1 brainsci-13-01339-t001:** Values of $n_{hk}$ for *T*, *A*, *O*.

*n*	*T*	*A*	*O*
$n_{00}$	7	8	7
$n_{01}$	4	3	4
$n_{10}$	2	2	8
$n_{11}$	9	9	3

**Table 2 brainsci-13-01339-t002:** Confidence intervals for *T*.

	Y=T
$I_{0}\left(Y\right)$	[285.79, 422.88]
$I_{1}\left(Y\right)$	(115.88,285.79]∪[422.88,1200.01)

**Table 3 brainsci-13-01339-t003:** Confidence intervals for *A*.

	Y=A
$I_{0}\left(Y\right)$	[14.62, 21.30]
$I_{1}\left(Y\right)$	[5.27,14.62]∪[21.30,35.89]

**Table 4 brainsci-13-01339-t004:** Confidence intervals for *O*.

	Y=O
$I_{0}\left(Y\right)$	[51.56, 75.18]
$I_{1}\left(Y\right)$	[43.48,51.56]∪[75.18,84.21]

**Table 5 brainsci-13-01339-t005:** Algebraic conditions on the least values of λ and μ for *T*, *A*, *O*.

	*T*	*A*	*O*
λ	5/7	3/4	0
μ	5/9	2/3	0

**Table 6 brainsci-13-01339-t006:** Values of parameters *T*, *A*, *O* and confidence intervals for S37.

	Value	Interval
*T*	207.85	$I_{1}$
*A*	8.66	$I_{1}$
*O*	48%	$I_{1}$

**Table 7 brainsci-13-01339-t007:** values of parameters *T*, *A*, *O* and confidence intervals for S08.

	Value	Interval
*T*	1119.51	$I_{1}$
*A*	33.92	$I_{1}$
*O*	47.06%	$I_{1}$

**Table 8 brainsci-13-01339-t008:** Values of parameters *T*, *A*, *O* and confidence intervals for S15.

	Value	Interval
*T*	289.03	$I_{0}$
*A*	12.57	$I_{1}$
*O*	58.33%	$I_{0}$

**Table 9 brainsci-13-01339-t009:** Average values and standard deviations for updated probabilities.

I			II			III			
$\mu_{h}$	$\mu_{d}$	σ	$\mu_{h}$	$\mu_{d}$	σ	$\mu_{h}$	$\mu_{d}$	σ	
0.30	0.70	0.16	0.36	0.64	0.10	0.36	0.64	0.07	**S37**
0.46	0.54	0.06	0.37	0.63	0.06	0.36	0.64	0.08	**S08**
0.37	0.63	0.16	0.46	0.54	0.13	0.49	0.51	0.13	**S15**

## Data Availability

Data available on demand at the Department of Psychology of University of Campania “L. Vanvitelli”: the person responsible is dr Anna Esposito.

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
