# Peer review of "A Dynamic Probabilistic Model for Heterogeneous Data Fusion: A Pilot Case Study from Computer-Aided Detection of Depression"

_brainsci, 2023, doi:10.3390/brainsci13091339_

Round 1

Reviewer 1 Report

The paper "A dynamic probabilistic model for heterogeneous data fusion: a pilot case study" presents a framework to identify detection based on computer-aided methods. The research was tested on 24 subjects. The results show that the designed solution had good outcomes.

- Originality/Novelty: The work is original, as it is a research based on heterogeneous data fusion on which a dynamic probabilistic model was applied. The proposed solution can be further used in the diagnosis of depression. 

- Significance: The results of the research are interpreted properly. The conclusions are justified and support by the results which are displayed in tables.

Please also include some graphical illustration for the obtained results. Write further descriptions and comparisions about the triggered values. 

All hypotheses were specified and the outcomes were carefully analysed. 

- Quality of Presentation: The article is written appropriately, respecting the logical succession of sections. Data and analyses are presented inside tables. Please add more figures to reflect the cases of interest when depression was identified.

Do not include references in the abstract.

Which operating system was used to run the program based on the interface from Figure 1? It seems to be Windows 7. The date which is displayed is 05/04/2014. Pretty old. When was the study performed? During which period? 

It is observed that the research was approved by an ethical committee in 2016. It would be interesting to compare the subjects before and after the COVID-19 pandemic. As well as in the case of students who studied online and then returned to university.

The title of the paper needs to be formatted. Capitalize nouns, pronouns, adjectives, verbs (including phrasal verbs), adverbs, and subordinate conjunctions. Lowercase articles (a, an, the), coordinating conjunctions, and prepositions (regardless of length).

The references section needs to be enriched. It must include more recent papers from the last 3 years. A lot of the cited papers are really old. What you can also study is the identification of depression based on an electroencephalogram headset.

Please provide the technical characteristics of the PC on which the analysis was done.

Please avoid the usage of the pronoun "we". Reformulate all such phrases in order to have a more formal approach.

Provide further details about the results obtained in the paper tables.

- Scientific Soundness: The study of the proposed dynamic probabilistic model is well documented and will be useful for further researches.

12% of the paper text was taken from the author's paper - Bruno Carbonaro. "Modeling epidemics by means of the stochastic description of complex systems", Computational and Mathematical Methods, 2021. Please do not leave the previous published text identical.

- Interest to the Readers: The conclusions will surely interest the readers of the Brain Sciences journal, and not only them, as depression is a topic of interest, modtly after the COVID-19 pandemic.  

- Overall Merit: The benefit to publish this paper consists in a good documented article regarding a new means to detect depression. 

- English Level: The level of English language is advanced. Through the entire paper, the language was appropriate and understandable, being easy to follow the flow since the beginning.

Please proofread again the whole paper.

Author Response

Open Review

ANSWER TO REVIEWER 1

We thank the reviewer for the very useful comments and suggestions allowing us to improve our paper. In the following her/his requests are in yellow and below are our answers to them.

(x) I would not like to sign my review report 

( ) I would like to sign my review report 

Quality of English Language

( ) I am not qualified to assess the quality of English in this paper 

( ) English very difficult to understand/incomprehensible 

( ) Extensive editing of English language required 

( ) Moderate editing of English language required 

(x) Minor editing of English language required 

( ) English language fine. No issues detected 

Yes

Can be improved

Must be improved

Not applicable

Does the introduction provide sufficient background and include all relevant references?

( )

(x)

( )

( )

Are all the cited references relevant to the research?

( )

(x)

( )

( )

Is the research design appropriate?

( )

(x)

( )

( )

Are the methods adequately described?

( )

(x)

( )

( )

Are the results clearly presented?

( )

(x)

( )

( )

Are the conclusions supported by the results?

( )

(x)

( )

( )

Comments and Suggestions for Authors

The paper "A dynamic probabilistic model for heterogeneous data fusion: a pilot case study" presents a framework to identify detection based on computer-aided methods. The research was tested on 24 subjects. The results show that the designed solution had good outcomes.

- Originality/Novelty: The work is original, as it is a research based on heterogeneous data fusion on which a dynamic probabilistic model was applied. The proposed solution can be further used in the diagnosis of depression. 

- Significance: The results of the research are interpreted properly. The conclusions are justified and support by the results which are displayed in tables.

Q1. Please also include some graphical illustration for the obtained results. Write further descriptions and comparisions about the triggered values. 

R. Done (we hope). We have added 4 figures that should clarify how our procedure modifies the probability that a subject is depressed with respect to that (certainty, both in the affirmative and in the negative case) assigned by the BDI.

All hypotheses were specified and the outcomes were carefully analysed. 

  • Quality of Presentation: The article is written appropriately, respecting the logical succession of sections. Data and analyses are presented inside tables. 

Q2. Please add more figures to reflect the cases of interest when depression was identified.

R. Figures 2 and 4, in particular, concern cases in which the subjects were recognized as depressed by the BDI

Q3. Do not include references in the abstract.

R. Done

Q4. Which operating system was used to run the program based on the interface from Figure 1? It seems to be Windows 7. The date which is displayed is 05/04/2014. Pretty old. When was the study performed? During which period? 

R. Done. The answers to these questions have been included in the Introduction to the paper.

“The recording software/hardware consisted of (as reported in page 4, line 8 of the paper, from the top) a clip-on microphone (Audio-Technica ATR3350), with external USB sound card. 

This software/hardware was developed inside the European Space Agency (ESA 2012) funded project (n. HSO-US 2012-108 )“Psychological Status Monitoring by Computerised Analysis of Language phenomena (COALA)” of which one of the authors (prof. Anna Esposito) was the coordinator.

It is a standard software/hardware equipped of a sound card to sample the speech data. It can run on any operating system It was installed on Window 7 for collecting the COALA data. In the following years  it has been installed on the operating systems windows 8, 9 and 10 and can be installed also on windows 11. The data exploited in this study were collected between 2016-2017, after getting the permission of the ethical committee of  the department of psychology of the Università della Campania “Luigi Vanvitelli”. The date displayed in figure 1 and the time in which the data were collected do not affect the originality and innovation  of the results presented in this study, proposing a mathematical model to fuse heterogeneous data.

Q5. It is observed that the research was approved by an ethical committee in 2016. It would be interesting to compare the subjects before and after the COVID-19 pandemic. As well as in the case of students who studied online and then returned to university.

R. We thank the referee for this suggestion, which had escaped our attention. Of course, it would be of the greatest interest to conduct an analysis of the depressive effects of COVID on a large sample, both through the BDI and speech analysis. Unfortunately, we do not have data for this circumstance,  but we do intend to investigate in this direction as soon as we will be able to collected data recorded during COVID and after. This remark has been also inserted in the last Section of the paper about research perspectives.

Q6. The title of the paper needs to be formatted. Capitalize nouns, pronouns, adjectives, verbs (including phrasal verbs), adverbs, and subordinate conjunctions. Lowercase articles (a, an, the), coordinating conjunctions, and prepositions (regardless of length).

R. Done. The title has been formatted and also changed as requested by reviewer 2. The new title of the paper is now: “A Dynamic Probabilistic Model for Heterogeneous Data Fusion: A Pilot Case Study from Computer-Aided Detection of Depression”  

Q7. The references section needs to be enriched. It must include more recent papers from the last 3 years. A lot of the cited papers are really old. What you can also study is the identification of depression based on an electroencephalogram headset. Some research regarding this is the following:

Iuliana Marin, Ioana-Andreea Dinescu, Teodora-Coralia Deleanu, Lujain Alshikh Sulaiman, Sarmad Monadel Sabree Al-Gayar, Nicolae Goga, Brain Performance Analysis based on an Electroencephalogram Headset, 12th International Conference on Electronics, Computers and Artificial Intelligence (ECAI), 25-27 June 2020, ISSN 2378-7147, WOS:000627393500105.

Iuliana Marin, Study of Mental Health and Learning Engagement during COVID-19 Pandemic based on an Electroencephalogram Headset, 13th annual International Conference of Education, Research and Innovation (ICERI2020), 9-10 November 2020.

R. Done. We inserted the references suggested by the review in the last Section of research perspectives. We also expanded the introduction providing more updated papers (reported below) on the automatic analysis of behavioral features for the detection of depression. 

Kanter, J.W., Busch, A.M., Weeks, C.E. et al. The nature of clinical depression: Symptoms, syndromes, and behavior analysis. BEHAV ANALYST 31, 1–21 (2008). https://doi.org/10.1007/BF03392158

Remes, O.; Mendes, J.F.; Templeton, P. Biological, Psychological, and Social Determinants of Depression: A Review of Recent Literature. Brain Sci. 2021, 11, 1633. https://doi.org/10.3390/brainsci11121633

Xue, D.; Guo, X.; Li, Y.; Sheng, Z.; Wang, L.; Liu, L.; Cao, J.; Liu, Y.; Lou, J.; Li, H.; et al. Risk Factor Analysis and a Predictive Model of Postoperative Depressive Symptoms in Elderly Patients Undergoing Video-Assisted Thoracoscopic Surgery. Brain Sci. 2023, 13, 646. https://doi.org/10.3390/brainsci13040646

Tao F; Ge X; Ma W; Esposito A, Vinciarelli A (2023) Multi-Local Attention for Speech-Based Depression Detection. In ICASSP 2023 - 2023 IEEE International Conference on Acoustics, Speech and Signal Processing (ICASSP), DOI: 10.1109/ICASSP49357.2023.10095757

https://ieeexplore.ieee.org/document/10095757

Lee Y-S, Park W-H. Diagnosis of depressive disorder model on facial expression based on fast R-CNN. Diagnostics. (2022) 12:317. doi: 10.3390/diagnostics12020317

Liu D, Liu B, Lin T, Liu G, Yang G, Qi D, Qiu Y, Lu Y, Yuan Q, Shuai SC, Li X, Liu O, Tang X, Shuai J, Cao Y and Lin H (2022) Measuring depression severity based on facial expression and body movement using deep convolutional neural network. Front. Psychiatry 13:1017064. doi: 10.3389/fpsyt.2022.1017064

Nolazco-Flores JA, Faundez-Zanuy M, Velázquez-Flores OA, Del-Valle-Soto C, Cordasco G, Esposito  A (2022) Mood State Detection in Handwritten Tasks Using PCA–mFCBF and Automated Machine Learning. Sensors 2022, 22(4), 1686; https://doi.org/10.3390/s22041686

Greco C, Matarazzo O,  Cordasco G, Vinciarelli A, Callejas Z, Esposito A (2021) "Discriminative Power of EEG-Based Biomarkers in Major Depressive Disorder: A Systematic Review," in IEEE Access, vol. 9, pp. 112850-112870, 2021, doi: 10.1109/ACCESS.2021.3103047, eid=2-s2.085112193360, ISSN: 21693536.

Aloshoban N, Esposito A, Vinciarelli A (2021) “What You Say or How You Say It? Depression Detection Through Joint Modelling of Linguistic and Acoustic Aspects of Speech“ Cognitive Computation, 1-14, DOI: 10.1007/s12559-020-09808-3, http://link.springer.com/article/10.1007/s12559-020-09808-3

Tao, F., Esposito, A., Vinciarelli, A. (2020) Spotting the Traces of Depression in Read Speech: An Approach Based on Computational Paralinguistics and Social Signal Processing. Proc. Interspeech 2020, 1828-1832, DOI: 10.21437/Interspeech.2020-2888.

Esposito A, Raimo G, Maldonato MN, Vogel C, Conson M, Cordasco G (2020) Behavioral Sentiment Analysis of Depressive States. 11th IEEE International Conference on Cognitive Infocommunications (CogInfoCom), Mariehamn, Finland, 2020, 209-214, doi: 10.1109/CogInfoCom50765.2020.9237856. https://ieeexplore.ieee.org/document/9237856, ISSN: 2380-7350

Likforman-Sulem L, Esposito A, Faundez-Zanuy M, Clémençon S, Cordasco G (2017) EMOTHAW: A Novel Database for Emotional State Recognition from Handwriting and Drawing. IEEE Transactions on Human-Machine Systems, 47(2):273-284, DOI: 10.1109/THMS.2016.2635441, http://ieeexplore.ieee.org/document/7807324/, WOS: 000396401600010

Q8. Please provide the technical characteristics of the PC on which the analysis was done.

The analyses can be made on any PC since they are not requiring any computational complexity. Particularly the PC  on wich the data were processed was a laptop ASUS.

Processing unit: Intel ® Core ™ i-7-6500U CPU, 2.50 GHz – 2.59 GHz

Installed RAM: 8,00 GB

Operating System windows 10 home version 22H2, 64 bit

Q9. Please avoid the usage of the pronoun "we". Reformulate all such phrases in order to have a more formal approach.

R. Done

Q10. Provide further details about the results obtained in the paper tables.

- Scientific Soundness: The study of the proposed dynamic probabilistic model is well documented and will be useful for further researches.

Q11. 12% of the paper text was taken from the author's paper - Bruno Carbonaro. "Modeling epidemics by means of the stochastic description of complex systems", Computational and Mathematical Methods, 2021. Please do not leave the previous published text identical.

R. Done. Done. Text has been changed to the extent it was possible, since some periods and  paragraphs report standard mathematical definitions, or conditions, or prescriptions and —  as usual in mathematical papers — cannot be stated with different words without altering their meaning. These had to be left unchanged. 

- Interest to the Readers: The conclusions will surely interest the readers of the Brain Sciences journal, and not only them, as depression is a topic of interest, modtly after the COVID-19 pandemic.  

- Overall Merit: The benefit to publish this paper consists in a good documented article regarding a new means to detect depression. 

- English Level: The level of English language is advanced. Through the entire paper, the language was appropriate and understandable, being easy to follow the flow since the beginning.

Comments on the Quality of English Language

Please proofread again the whole paper.

Submission Date

24 July 2023

Date of this review

31 Jul 2023 08:46:21

© 1996-2023 MDPI (Basel, Switzerland) unless otherwise stated

Disclaimer Terms and Conditions Privacy Policy

Reviewer 2 Report

The author has explored the application of computer-aided depression detection. By analyzing diverse experimental data and their correlations, the author aims to develop a unified approach that enhances the accuracy of diagnosing depression by incorporating statistical parameters from speech analysis to modify diagnoses provided by the Beck Depression Inventory (BDI-II). The proposed model has valuable potential contributions to mental health assessment.

Here are further suggestions for the authors to consider:

Further discussion is needed on how the model tackles complex depression challenges with consideration of more personalized measurement.

More details are required on applying the model to speech data and translating speech features into probabilistic distributions. 

The title should be more informative about the depression case study.

The generalization ability of the method to other disorders, may beed to be further discussed. Such as, a review on related data sets, and point our future research directions.

Author Response

Open Review

( ) I would not like to sign my review report 

(x) I would like to sign my review report 

Quality of English Language

( ) I am not qualified to assess the quality of English in this paper 

( ) English very difficult to understand/incomprehensible 

( ) Extensive editing of English language required 

( ) Moderate editing of English language required 

( ) Minor editing of English language required 

(x) English language fine. No issues detected 

Yes

Can be improved

Must be improved

Not applicable

Does the introduction provide sufficient background and include all relevant references?

(x)

( )

( )

( )

Are all the cited references relevant to the research?

(x)

( )

( )

( )

Is the research design appropriate?

(x)

( )

( )

( )

Are the methods adequately described?

(x)

( )

( )

( )

Are the results clearly presented?

(x)

( )

( )

( )

Are the conclusions supported by the results?

(x)

( )

( )

( )

Comments and Suggestions for Authors

The author has explored the application of computer-aided depression detection. By analyzing diverse experimental data and their correlations, the author aims to develop a unified approach that enhances the accuracy of diagnosing depression by incorporating statistical parameters from speech analysis to modify diagnoses provided by the Beck Depression Inventory (BDI-II). The proposed model has valuable potential contributions to mental health assessment.

Here are further suggestions for the authors to consider:

Q1. Further discussion is needed on how the model tackles complex depression challenges with consideration of more personalized measurement.

R. Done. A short account of this point is given in the Introduction.

Q2. More details are required on applying the model to speech data and translating speech features into probabilistic distributions. 

R. The whole paper, in particular Sections 3, 4 and 5, is devoted to show how speech features can be translated into probabilistic distribution. Of course, the treatment is quite mathematical, and hard to be followed. In short, the (conditional) relative frequencies of the possible results of each test are interpreted as probabilities. This point is carefully described in Section 5.

Q3. The title should be more informative about the depression case study.

R. Done

Q4. The generalization ability of the method to other disorders, may beed to be further discussed. Such as, a review on related data sets, and point our future research directions.

R. Done. In the last Section on research persopectives. A review of data sets about other disorders would have required another paper, but we give a reference to a general treatment showing the independence of the method of the particular disease (that is, of the particular classification problem).

Submission Date

24 July 2023

Date of this review

08 Aug 2023 08:20:42

© 1996-2023 MDPI (Basel, Switzerland) unless otherwise stated

Round 2

Reviewer 1 Report

I approve the updated version of the article. The authors have modified the paper according to the received comments.